# Design of Functional RGD Peptide-Based Biomaterials for Tissue Engineering

**DOI:** 10.3390/pharmaceutics15020345

**Published:** 2023-01-19

**Authors:** Vijay Bhooshan Kumar, Om Shanker Tiwari, Gal Finkelstein-Zuta, Sigal Rencus-Lazar, Ehud Gazit

**Affiliations:** 1The Shmunis School of Biomedicine and Cancer Research, George S. Wise Faculty of Life Sciences, Tel Aviv University, Tel Aviv 6997801, Israel; 2Department of Materials Science and Engineering, The Iby and Aladar Fleischman Faculty of Engineering, Tel Aviv University, Tel Aviv 6997801, Israel; 3Sagol School of Neuroscience, Tel Aviv University, Ramat Aviv, Tel Aviv 6997801, Israel

**Keywords:** RGD, peptides, nanomaterials, biomaterials, tissue engineering

## Abstract

Tissue engineering (TE) is a rapidly expanding field aimed at restoring or replacing damaged tissues. In spite of significant advancements, the implementation of TE technologies requires the development of novel, highly biocompatible three-dimensional tissue structures. In this regard, the use of peptide self-assembly is an effective method for developing various tissue structures and surface functionalities. Specifically, the arginine–glycine–aspartic acid (RGD) family of peptides is known to be the most prominent ligand for extracellular integrin receptors. Due to their specific expression patterns in various human tissues and their tight association with various pathophysiological conditions, RGD peptides are suitable targets for tissue regeneration and treatment as well as organ replacement. Therefore, RGD-based ligands have been widely used in biomedical research. This review article summarizes the progress made in the application of RGD for tissue and organ development. Furthermore, we examine the effect of RGD peptide structure and sequence on the efficacy of TE in clinical and preclinical studies. Additionally, we outline the recent advancement in the use of RGD functionalized biomaterials for the regeneration of various tissues, including corneal repair, artificial neovascularization, and bone TE.

## 1. Introduction

The tissue level of organization is one of the most fundamental in the human body. Certain tissues have the remarkable ability to regenerate, making them useful as a treatment for a wide range of injuries, such as fractured bones, cuts in muscles, and damage to organs that are otherwise difficult to treat [1,2]. The regeneration of tissues is not always successful in the case of diseases and injuries, so surgical strategies are often required. Furthermore, the treatment and repair of extensive tissue damage may require tissue/cell grafts, replacements, or implants [1]. For this purpose, several strategies have been developed, such as autogenic and allogenic tissue transplantation [3]. Although transplants have several benefits, there are some drawbacks, including insufficient organ donors and lifelong immunosuppression [3]. Furthermore, transplanted organs often cannot function as native ones [4,5]. A variety of synthetic and natural materials have been utilized for tissue engineering (TE) and tissue implants aiming to address these issues. Importantly, TE is not only intended to restore damaged tissues but also to assemble functional constructs that promote regeneration and enhance cell viability and migration. Therefore, an implantable material must be biocompatible, biodegradable, and have proper functional sites that can adapt to biodegradation. Importantly, biomedical materials must be able to interact with cells in vivo to avoid any potential foreign body reactions such as implant encapsulation, thrombosis, embolization, and aseptic loosening. Unlike metals and ceramics, which are non-biodegradable and difficult to process, peptides show high design flexibility due to their structure and composition, allowing their facile design to support cell viability and proliferation [6,7,8]. 

Nanomaterial-based TE scaffolds serve as a template for cell interactions and the formation of the extracellular matrix (ECM), which forms a sustainable assembly [9]. Additionally, such scaffolds provide structural support for newly formed tissues [10,11]. Cell colonization, migration, and differentiation, as well as tissue growth, can be supported by peptide-based materials, which often control the development of the tissue [12]. A peptide-based scaffold should comprise interconnected porosities, surface functional groups to allow cell interactions, and adequate mechanical properties. The optimal scaffold should also be biodegradable or bioresorbable, nontoxic upon degradation, and easy to construct [12]. The tremendous advancements in this field in recent years emphasize its potential to revolutionize biomedical research. Recently, researchers have reported that to biologically activate biomaterial surfaces, the arginine–glycine–aspartic acid (RGD) sequence was immobilized on biomaterial surfaces in order to stimulate cell adhesion through specific interactions with integrin receptors [13]. The binding of RGD to integrin receptors plays a major role in the cell adhesion mechanism. RGD has been further shown to improve cell density and migration in the microenvironment. In tissue/cell implants, for example, it does not only increase tissue growth but also decreases interface fibrous tissue [14,15,16]. Promising results were obtained when RGD was conjugated with natural polymers such as hyaluronic acid and collagen [17]. 

Here, we present several different RGDs and RGD-based materials that can significantly influence the binding of cells or tissues to the surface of biomaterials. We also highlight the application of various RGD and functional RGD motifs in various multidisciplinary areas, including tissue growth, neurovascular regeneration, corneal repair, artificial neovascularization, and bone TE. Finally, we summarize the clinically relevant properties of RGD peptides that have been used in clinical or preclinical trials.

## 2. Biophysico-Chemical and Biological Properties of RGD for TE

The ability of cells to adhere to biomaterials is an important characteristic of selective cell retention (SCR) technology, which is used to rapidly fabricate new tissue grafts in clinical settings [18]. The main approach for improving the adhesive properties and regeneration of tissues focuses on bio-mimicry by physical and biological manipulations. Several important ECM molecules contain the several RGD sequence, and the artificial cyclic RGD (cRGD, c[RGDfK]) peptides have been shown to maintain specific cell adhesion and promote the regeneration of tissue [13,18,19]. Since SCR technology involves a real-time process of removing bone marrow cells, it requires an RGD sequence that demonstrates both specific adhesion as well as rapid, nonspecific adhesion according to the requirement, thus accelerating the process [17,20]. Positive charges play an important role in the promotion of nonspecific cell adhesion [17]. Luo et al. have reported a study of 20 lysine sequences (20K) located on the lateral chain of the lysine in a cRGD peptide [21]. This lysine-cyclic RGD (LcRGD, c[RGDfK]-20K) peptide was designed to coordinate the rapid nonspecific adhesion force mediated by the positive charge and the specific interaction between the RGD sequence and integrins to enable the system to retain cells, especially osteogenic progenitor cells, in a manner compatible with SCR technology (Figure 1) [21]. The use of functionalized RGD materials with short cell-adhesion peptides (CAPs) has led to a better understanding of the role of the extracellular matrix. It is possible to easily translate RGD materials that have been functionalized with CAP into robust 3D cell culture systems for tissue engineering applications. Unfortunately, the panel of CAPs available for designing cell culture environments is limited to non-modified RGDs. A high-throughput screening approach has been proposed in order to evaluate the role of CAPs as well as identify their precise impact on cell adhesion. In spite of the fact that microarray screening platforms have been established for studying cell adhesion, high-throughput screening has yet to be applied to tissue engineering and 3D cell culture. It is important to understand the role of CAPs in tissue engineering and design, not only for their cell-adhesion properties but also to drive cellular functions and organization.

It has been proposed that RGD can also serve as a biomimetic peptide that promotes cell adhesion to the matrix, prevents cell apoptosis, and promotes the regeneration of new tissue that is attached to the matrix [13]. Several researchers have also designed TE matrices aimed at regenerating damaged myocardial tissues using heart patches and other heart tissue [22,23]. RGD-immobilized or unmodified alginate scaffolds were used to seed neonatal rat cardiac cells. In addition to improving cell adhesion to the matrix, the immobilized RGD peptide prevented cell apoptosis and accelerated cardiac tissue regeneration [24]. The images of α-actinin-stained cells shown in Figure 2 indicate that there is an additional population of nonmyocyte cells (NMCs), which are negatively stained for α-actinin, but whose nuclei are stained red by propidium iodide [25]. The NMCs are arranged around the stretched myocardial fiber in RGD–alginate scaffolds on day 12 of cultivation (Figure 2A), similar to the native, mature ventricular cardiac tissue (Figure 2C) [25]. Figure 2B, on the other hand, shows an unordered structure formed on an alginate-alone scaffold. Immunostaining for α-actinin (green) followed by confocal microscopy imaging showed organized myofibrils within the cardiomyocyte culture grown on the RGD-functionalized alginate scaffolds. Additionally, adjacent cardiomyocytes joined together to form striated myofibers (Figure 2D, day 6), which became more frequent as cultivation progressed (Figure 2E, day 12) [25]. The cardiomyocytes cultured on unmodified alginate scaffolds, however, demonstrated randomized myofibrils, interactions between adjacent cardiomyocytes were less frequent, and myofibers were not detected (Figure 2F,G, days 6 and 12, respectively). On day 10 of culture, immunostaining for troponin-T (Tn-T) revealed striated morphology of cardiomyocytes in the two construct types. Myofibers were formed in the RGD-functionalized alginate scaffolds (Figure 2H) but not in the non-functionalized scaffolds (Figure 2I) [25]. The authors also used Western blotting to determine α-actinin levels in order to examine the sarcomeric stage of organization. However, the α-actinin levels decreased over time in both the RGD–alginate scaffold and the alginate-alone scaffold **(**Figure 2J). In the later cultivation times, on days 8 and 12, the differences reached statistical significance. When testing the N-Cadherin protein level, an intercalated disk protein of cardiomyocytes, a significant change was detected from the fourth day onwards, showing higher levels in the RGD-immobilized alginate scaffold (Figure 2K). On the second day, the levels of the major gap junctional protein connexin-43, which is expressed on cardiomyocytes and NMCs, were also significantly increased in both cultures but more prominently in the RGD–alginate scaffolds compared to the control unmodified alginate scaffolds, indicating a clear preference for the RGD–alginate scaffold as a cultivation system for cardiac cells (Figure 2L) [25]. After six days, cardiomyocytes reorganized their myofibrils and reconstructed myofibers composed of multiple cardiomyocytes in a typical bundle of myofibers. As a result of adhering to the RGD-alginate matrix, the NMC population, primarily cardiac fibroblasts, played a vital role in supporting the cardiomyocytes and was mostly observed to surround bundles of cardiac myofibers in a manner similar to that of native cardiac tissue [25]. Taken together, RGD peptide immobilization into macro-porous alginate scaffolds significantly promotes cardiac TE, contributing to the formation of functional cardiac muscle tissue and improving the preservation of regenerated tissue.

Li et al. developed RGD-modified graphene oxide (GO) nanomaterials by covalently linking the carboxyl groups on the surface of GO with the amine groups of the RGD peptide [26]. Through this synthesis procedure, GO is not only endowed with desirable biocompatibility but also maintains its physical and chemical properties. Figure 3A shows the procedure for modifying the RGD peptide onto the GO surface and the results of the cell analysis experiment. GO films were first formed by the natural drying of an aqueous solution. RGD peptides with amine functional groups were then immobilized on GO film surfaces via the EDC/NHS coupling amidation reaction [26]. Using human periodontal ligament fibroblasts (HPLFs) as a model, the RGD-GO films were investigated for their effects on adhesion and proliferation, along with unmodified GO controls. RGO-GO-modified nanomaterials were deposited onto the electrode surface in order to study how their electrochemical impedance spectroscopy changed following HPLF culturing with and without chemical treatment to further investigate their potential applications in cytological analysis in vitro. Further studies were conducted to confirm the biocompatibility of the formed films [26]. Cells spread evenly on both films and displayed well-structured cytoskeletons (Figure 3B) with the modification of the RGD peptide resulting in significantly improved adhesion and proliferation. It may therefore be possible to use RGD-GO nanomaterial for applications other than dental TE. 

Biological tissues fabricated in vitro could be a valuable tool to screen newly synthesized biodegradable RGD peptides to gain a deeper understanding of tissue development [27,28]. Controlling the growth and morphology of engineered tissues, however, remains a challenge. In order to modulate tissue growth, Taketa et al. reported several new techniques [29]. In order to demonstrate the effect of RGD on epithelial tissue growth, epithelial tissue was isolated from whole submandibular gland (SMG) tissue and cultured on RGD-modified alginate hydrogel sheets [29]. Epithelial cells cultured exclusively on RGD-modified alginate hydrogels did not grow but instead degenerated. RGD did not affect the expression of matrix proteins in SMG tissues, and both fibroblast growth factor 7 (FGF7) and FGF10 were predominant in mesenchymal tissues (Figure 4A). Western blot analyses revealed a significant increase in both growth factors in SMG cultured for 72 h on RGD-modified gels (Figure 4B). In order to confirm the roles of these growth factors, antibodies toward FGF7 and FGF10 were applied to the SMG cultured on the RGD-treated gel sheet (Figure 4A). As shown in Figure 4C, the SMG cultured on the RGD-modified gel sheet showed a characteristic distribution of neural tissue, whereas the SMG cultured on the unmodified gel sheet showed non-growing, localized neuronal tissue (Figure 4C) [29]. The effect of RGD on neurite outgrowth was confirmed by cultivating PC12 cells on alginate hydrogel sheets with or without RGD modification for 7 days. The neuronal growth of PC12 cells was significantly accelerated when cultured on a substrate containing RGD (Figure 4D,E). Additionally, antibody toward neurturin (NRTN), a neurotrophic factor, was used in SMG cultures grown on RGD-modified gel sheets. Strikingly, the growth of SMG was attenuated when cultured on an RGD-modified hydrogel sheet in the presence of the anti-NRTN antibodies (Figure 4F) [29]. Furthermore, small RGD-modified alginate beads were placed on growing SMG tissue. The RGD-modified beads successfully induced cleft formation at the bead position, guiding the desired SMG morphology [29]. Consequently, this RGD-modified material may be a promising tool for modulating tissue growth and morphology in vitro.

The development of TE scaffolds for promoting cellular growth/proliferation has received considerable research attention in recent years. Shin et al. examined the potential of RGD peptides and GO co-functionalized poly (lactide-co-glycolide, PLGA) nanofiber mats to serve as biofunctional scaffolds for vascular TE by characterizing their physicochemical properties [14]. The RGD-GO-PLGA nanofiber mats were readily fabricated and consisted of randomly oriented electrospun nanofibers with an average diameter of 558 nm as determined by scanning electron microscopy. RGD peptide and GO were co-functionalized into PLGA nanofibers, as confirmed by Fourier-transform infrared spectroscopy. Furthermore, the surface hydrophilicity of the nanofiber mats was significantly increased by co-functionalization with the RGD peptide and GO [14]. Under the conditions of cell culture, the mats were found to be thermally stable. In addition, immunofluorescence staining was used to evaluate the adherence and fast differentiation of vascular smooth muscle cells (VSMCs) on the RGD-functionalized GO-PLGA nanofiber (Figure 5). The morphology of VSMCs grown on the PLGA nanofiber mat differed from those grown on GO-functionalized PLGA, RGD-functionalized PLGA, or RGD-functionalized GO-PLGA nanofiber mats (Figure 5A). The complete cell surface area of VSMCs cultured on RGD-functionalized GO-PLGA nanofiber mats significantly increased compared to those cultured on the other nanofiber mats, possibly due to the synergistic effect of the RGD peptide and GO (Figure 5B). Figure 5C displays the single-cell area of VSMCs on each nanofiber mat [14]. The results of these studies indicated that the RGD-GO-PLGA nanofiber mats could effectively promote the growth of VSMCs and may prove to be promising candidates for TE.

Shin et al. synthesized nanofiber matrices using mammalian M13 bacteriophage-displayed RGD peptides (RGD-M13 phage) and PLGA and characterized their mechanical and physicochemical properties (Figure 6A,B) [30]. The matrices represent a continuous and 3D network construction with interrelated pores that are very close to the natural ECM. Electrospun fibers composed of PLGA and RGD/PLGA were measured to be 1400 × 150 nm and 100 × 30 nm, respectively. The RGD-M13 phage showed green fluorescence along the RGD/PLGA nanofibers, as shown in Figure 6C [30]. The PLGA fiber, however, did not exhibit any fluorescence. The images in Figure 6D–F depict the morphologies of C2C12 myoblasts after 3 and 7 days on PLGA and RGD/PLGA matrices. On the PLGA matrix, the C2C12 myoblasts were unable to properly form an F-actin network, and their numbers did not increase significantly (Figure 6D). Conversely, the C2C12 myoblasts grown on the RGD/PLGA matrix exhibited a spindle-like morphology with well-organized F-actins and significantly increased cell numbers (Figure 6E) [30]. The stimulation of cell adherence and growth of myoblasts was shown by quantitative measurement of cell morphology (Figure 6F). Moreover, the effect of GO functionalization on the cellular activity of C2C12 myoblasts system on functionalized PLGA on RGD-M13 (RGD/PLGA) nanofiber matrices was demonstrated. These findings suggest that the combination of RGD/PLGA nanofiber matrices and GO can be used to engineer and regenerate skeletal tissues.

## 3. Applications of RGD-Based Materials

RGD is a tripeptide sequence, and the latter is a functional peptide containing RGD. Various RGD peptide sequences and hydrogels with varying structures have been used in tissue engineering for tissue regeneration. An RGD peptide can be classified as linear or circular. It is interesting to note that cyclic RGD peptides are believed to be more active than linear RGD peptides. Cyclic peptides likely have a higher affinity for integrin receptors and are more resistant to proteolysis. Additionally, Li et al. have study found that cyclic RGD was more beneficial to bone repair in vivo than linear RGD [31]. The modification of poly-carbonate urethane with a photoactive peptide sequence containing RGD also resulted in improved vascular grafts [32,33,34]. Moreover, cell adhesion depends on mechanical forces, growth factors, and the extracellular matrix [35]. To determine cell fate or cell adhesion, several physical, chemical, and functional relationships exist between a cell, growth factors, and the structure and rigidity of the surrounding matrix [35,36,37]. The interaction between EGFR (epidermal growth factor receptor) and integrins has been demonstrated in studies; however, most of the observed outcomes have been attributed to molecules downstream of the receptors, away from the plasma membrane [36,37]. In a recent study, Rao et al. used tension gauge tether probes to display the integrin ligand cRGDfK and to measure the tension between the integrins [38]. The results of this study suggest that EGFR regulates integrin tension as well as the spatial organization of focal adhesions [38]. Additionally, the researchers found that the mechanical tension threshold required for outside-in integrin activation can be adjusted by ligand-dependent EGFR signaling.

### 3.1. RGD Peptides to Enhance Neuronal Growth

Nerve TE is a rapidly expanding area of research that provides a novel and promising approach to nerve repair and regeneration [39,40,41]. For this purpose, biomaterials capable of providing a continuous path for regeneration, promoting the infiltration of cells and the release of inductive factors that promote axonal extension, are continuously developed [42]. In nerve regeneration studies, soft hydrogels are commonly infused with ECM proteins or fragments, as well as linear and cyclic peptides [43,44,45]. A thorough understanding of neuronal mechanisms and cell behavior in contact with different biomaterials is also necessary for the development of advanced prosthetic devices. RGD is a well-studied example due to its use as an integrin-mediated cell adhesion agent [46,47]. Vedaraman et al. investigated the use of short bicyclic peptides with RGD in a cyclic loop as a biochemical cue for cell growth inside three-dimensional synthetic poly(ethylene glycol) (PEG)-based anisogels [48]. Defining these peptides allows them to be highly selective in terms of either the bound integrin subunit, namely αvβ3 or α5β1. It is possible to formulate an aECM (artificial extracellular matrix) hydrogel that supports nerve growth through enzymatic conjugation of such bicyclic peptides to the PEG backbone. The authors also examined nerve cell migration and proliferation, resulting in enhanced cell growth using degradable peptide crosslinkers [48]. Mouse fibroblasts and primary nerve cells from embryonic chick dorsal root ganglions show superior growth in bicyclic RGD peptide conjugated gels selective towards αvβ3 or α5β1, compared to monocyclic or linear RGD peptides, with a slight preference to αvβ3 selective bicyclic peptides in the case of nerve growth [48]. As a result of this study, PEG hydrogels coupled with bicyclic RGD peptides can be utilized as an aECM model and pave the way for the development of integrin-selective biomolecules to promote cell growth and nerve regeneration.

### 3.2. Bone Tissue Engineering

The complex emerging field of bone TE encompasses osteoprogenitor cell biology as well as biomaterials [49,50,51,52]. Rachmiel et al. used electrospinning to fabricate a composite scaffold with a core/shell morphology composed of the polycaprolactone (PCL) polymer, hyaluronic acid, and fluorenylmethoxycarbonyl-RGD (Fmoc-FRGD) [53]. Electron microscopy imaging demonstrated that the fibrous network of the scaffold resembled the ECM structure and in vitro assays showed the composite scaffold could support pre-osteoblasts adhesion, proliferation, and differentiation of bone [53]. Alkaline phosphatase (ALP) activity plays a crucial role in bone mineralization and serves as a marker for bone formation. The increased ALP levels in cells cultured on HA-FmocFRGD-PCL core/shell fibers demonstrated the osteogenic potential of the scaffold [53]. Therefore, HA and FmocFRGD-based materials can promote cellular differentiation and facilitate biomineralization.

Research has been conducted to examine whether titanium (Ti) or cobalt–chromium alloy implants coated with RGD can improve osteogenic cell homing, progenitor cell differentiation, and osseointegration [54,55]. Due to the numerous advantages of Ti, such as biocompatibility, mechanical strength, and corrosion resistance, it is extensively used in the clinic for permanent implantation. To further improve the currently available Ti implants, surface processing methods should be developed to allow strong interactions between the implant surface and the bone cells [55]. The coating of Ti surfaces with bioactive materials, such as RGD peptide, is one of the most advanced techniques. As a result, bone healing can be greatly enhanced by improving the attachment and differentiation of osteoblast cells [55,56,57,58]. Accordingly, Amit et al. and other researchers evaluated the levels of osseointegration during in vivo healing and bone regeneration using a combination of biocompatible materials such as Ti-based materials, laser-grooved surfaces, and RGD coating alloys [59,60]. Therefore, new RGD-based materials or alloys modified with RGD peptide and phosphoserine can be developed to enhance osteogenesis and promote bone TE.

### 3.3. Cardiovascular Tissue

The development of cardiac TE is a promising method to cure cardiovascular diseases, which comprise a main socio-economic problem. Moreover, cardiovascular stiffness changes with age, which contributes to the onset and progression of diseases. The expression and cross-linking of collagen affect the rigidity of the cardiac ECM. Therefore, small-diameter (6 mm inner diameter) vascular grafts may serve as a basis for long-term treatment [61]. The field of vascular TE is a relatively new one that has undergone replacement with bio-functionalities [62]. RGD-modified peptide grafts exhibit the ability to inhibit platelet adhesion, resulting in enhanced cellular infiltration, endothelialization, and ECM formation [61,63]. This is a promising approach to improve the biocompatibility of small-caliber vascular grafts, but it is unclear what impact different sequence compositions will have on graft performance [64]. Microporous PCL scaffolds functionalized with RGD, which significantly increase endothelial cell adhesion rates, may be an excellent choice for the clinical use of small-diameter vascular grafts [65]. Additionally, a macro-porous scaffold containing RGD-alginate was used to enhance the regeneration and preservation of cardiac tissue. Using RGD to modify alginate scaffolds, cardiac cell morphology changed from compacted clusters to elongated and outspread structures. As a result, more collagen was deposited, and cellular apoptosis was successful. The successful regeneration of cardiac cells on the RGD scaffold was concluded to be due to the appropriate kinetics of cell adherence when compared to the unmodified scaffold [25]. According to Hawkes et al., the nanoscale adhesion organization, signaling, and traction force generation of neonatal rat cardiomyocytes (which express both laminin-binding and fibronectin-binding integrins) are strongly dependent on the combination of integrins and their ligands [66]. Using an electrospinning technique, Zhu et al. developed poly(ester-urethane) urea nanoscale fibers and modified them with an RGD peptide to produce a new scaffold for the engineering of vascular tissues [67]. Covalently coupling Ac-GRGD (acrylamide-terminated glycine-arginine-glycine-aspartic) peptide to PEUU (poly(ester-urethane) urea) nanofibrous mats promoted cell attachment and fast differentiation. The Ac-GRGD-modified PEUU nanofibrous mats displayed high cytocompatibility towards human umbilical vein endothelial cells (HUVECs) as well as high compatibility towards rabbit platelet-rich plasma and red blood cells during hemocompatibility and biocompatibility analysis [67]. A new RGD-GO PLGA nanofiber scaffold was developed by Shin et al. using electrospinning and co-functionalized with RGD peptide and GO [14]. RGD-PLGA and RGD-GO-PLGA nanofibers were found to better attach to VSMCs than PLGA and GO-PLGA nanofibers [14]. Furthermore, immunofluorescence staining confirmed that RGD-PLGA and RGD-GO-PLGA mats contained a higher number of spindle-shaped VSMCs with well-developed F-actin than PLGA and GO-PLGA mats [14]. Kramer et al. recently examined the interaction of neonatal rat heart cells with engineered spider silk proteins (eADF4(C16)) tagged with the RGD peptide, which can be 3D-printed or applied as a coating. Based on the results, eADF4(C16)-RGD may be a promising material for cardiac TE. Additionally, it is essential to develop a fabrication technique that will permit the design of hierarchical structures and constructs with high densities of cardiomyocytes for optimal contractility. Therefore, the emphasis is on the identification of materials that are suitable for the engineering of cardiac tissue.

### 3.4. Cornea Repair

Millions of people worldwide suffer from corneal blindness, one of the most common causes of vision loss, either due to aging or various diseases [68]. Several approaches have been explored to engineer biomaterials that are suitable for corneal transplantation in order to treat these conditions. The incorporation of bioactive molecules, as well as different combinations of materials and fiber crosslinking methods, have been investigated [69,70,71]. Using RGD to modify a surface improves cell attachment, alignment, proliferation, and expression of collagen types I and V, as well as proteoglycan types [72]. A study by Nili et al. demonstrated that recombinant RGD-silk fibroin could be enhanced by RGD surface modification, and human corneal limbal epithelial cells grow better when the corneal stroma tissue is integrated with proteoglycan-rich ECM [70]. Generating hierarchical structures in corneal stromal tissue and replicating their patterns is a potential strategy for engineering human corneas. Furthermore, this technique, as well as other tissue-repairing techniques, may be underlined by complex mechanisms that require more exploration [69,70,71]. A three-dimensional culture of stem cells in media supplemented with an RGD-alginate gel increased cell viability, whereas, for a stem cell-derived embryonic body, alginate without RGD caused dissociation and an abnormal cyclic structure [73,74]. The formation of the retinal pigmented epithelium (RPE) from stem cells in the presence of scaffolds containing RGD alginate has been investigated [75]. Based on these results, RGD-alginate scaffolds offer preferential advantages over current hyaluronic acid-based hydrogels for retinal cell delivery [74].

## 4. RGD-Based Materials for Different Types of Tissue Engineering Applications

The RGD motif is a suitable choice for probes that can be used for tissue repair, diagnosis, theranostics, and TE since it is easy to combine with integrins expressed on tumor vessels. Despite some disadvantages, immune RGD-based systems should be evaluated by new approaches if they are to achieve further clinical acceptance. The pre-clinical applications and clinical trials of RGD-modified systems for different applications [62] are briefly presented in Table 1.

## 5. Challenges, Conclusions, and Future Perspectives on RGD-Based Materials for TE

Many novel discoveries have led to advancements in TE and provided new insights into the nature of different peptides or biomaterials that can improve TE applications. Several factors contribute to the extensive use of peptides in biomedical applications, including their ease of chemical synthesis, small size, high stability, and low immunogenicity. Specifically, RGD is one of the most common examples of peptide-based materials for TE. The development of new RGD peptide biomaterials highlights these scaffolds as promising candidates for physical and biomedical applications. The purpose of this review is to provide an overview of the most recent research findings on RGD-based systems in tissue regeneration and TE. Due to its high affinity for integrin receptors, the functionalized RGD peptide enhances the specific attachment of biologically derived nanoscale materials to animal cells/tissue expressing these receptors. The RGD peptide has been shown to be effective in a number of in vitro studies; however, some in vivo studies utilizing animal models did not show a significant correlation with the in vitro results [57]. RGD efficacy could be affected by a wide range of factors, such as the density and structure of RGD motifs and the physiochemical properties of the biomaterials, potentially explaining these seemingly contradictory results. It should be noted, however, that a considerable amount of preclinical research showed the high effectiveness of RGD peptides in specifically binding to cells expressing integrin receptors for a variety of applications. Several researchers are developing biomaterials based on RGD for implantation into organs. RGD-based materials can be used as replacement parts or to stimulate the growth of natural tissue. The future development of functionalized RGD replicas of human tissues or organs will provide scientists with the capability not only to study the effects of novel drugs or treatments but also to communicate with these replicas, allowing sensing and monitoring of the healing process. Accordingly, it is expected that RGD peptides will be used in TE applications in the near future, thereby making significant progress in preclinical and clinical advancements.

## Figures and Tables

**Figure 1 pharmaceutics-15-00345-f001:**
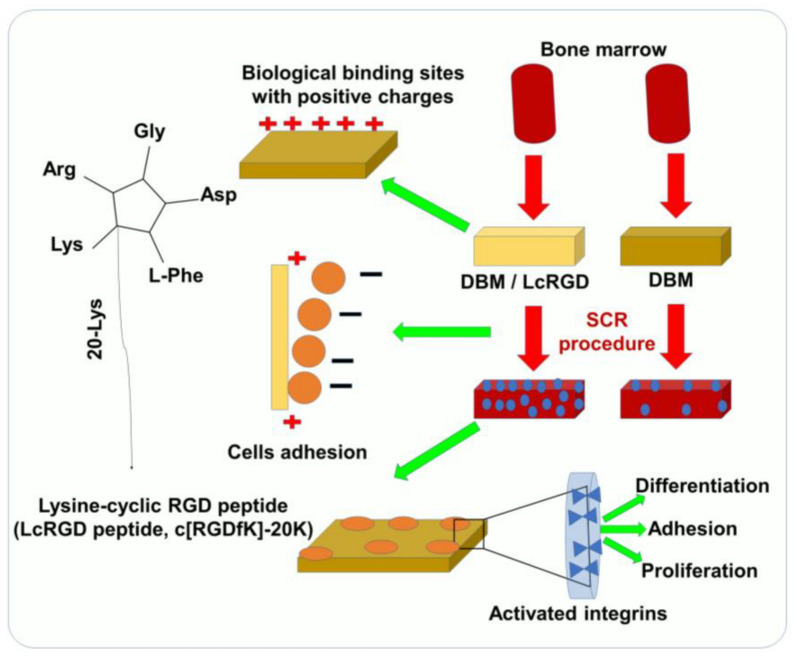
Using SCR technology for rapid TE and bone construction, LcRGD-peptide-modified demineralized bone matrix (DBM) exhibits a superior ability to adhere to cell surfaces and promote osteoinduction.

**Figure 2 pharmaceutics-15-00345-f002:**
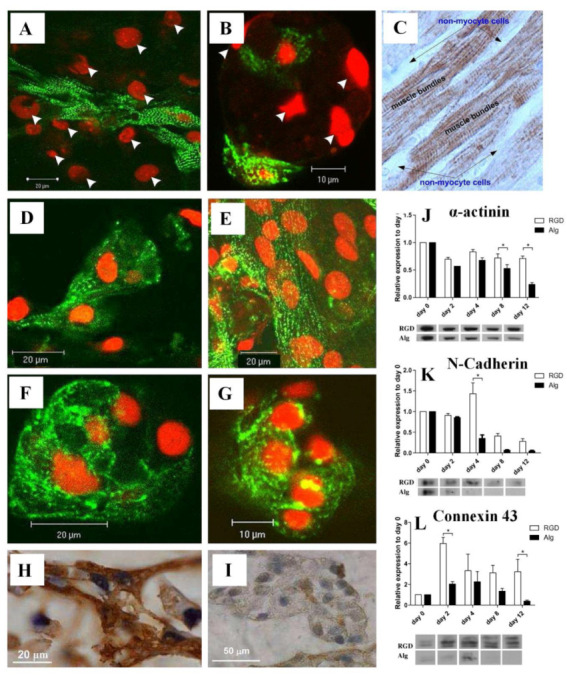
(**A**,**B**) Relative locations of cardiomyocytes and NMCs in (**A**) RGD-immobilized alginate scaffold and (**B**) unmodified alginate scaffold. Cardiomyocytes were stained for α-actinin (green), while nuclei were stained with PI (red). The arrowheads indicate NMC nuclei. (**C**) A native adult cardiac tissue (brown) stained for troponin-T. A negative staining of the NMCs surrounding the cardiomyocyte bundles can be detected. (**D**–**G**) Confocal microscopy images of immune-stained cardiac cells grown on (**D**,**E**) RGD-immobilized scaffolds and (**F**,**G**) unmodified alginate scaffolds for (**D**,**F**) six and (**E**,**G**) twelve days. (**H**,**I**) Representative cardiomyocyte morphologies as revealed by immunostaining of paraffin-embedded sections on day 10 of culture for Tn-T in (**H**) the RGD–alginate or (**I**) unmodified scaffolds. (**J**–**L**) Western blot analysis of cardiomyocytes and nonmyocytes using antibodies towards (**J**) α-actinin (* *p* = 0.0009), (**K**) N-cadherin (* *p* = 0.0001), and (**L**) connexin-43 (* *p* = 0.0525). Reproduced with permission from Ref. [25]. Copyright © 2010 Acta Materialia Inc. Published by Elsevier Ltd. All rights reserved.

**Figure 3 pharmaceutics-15-00345-f003:**
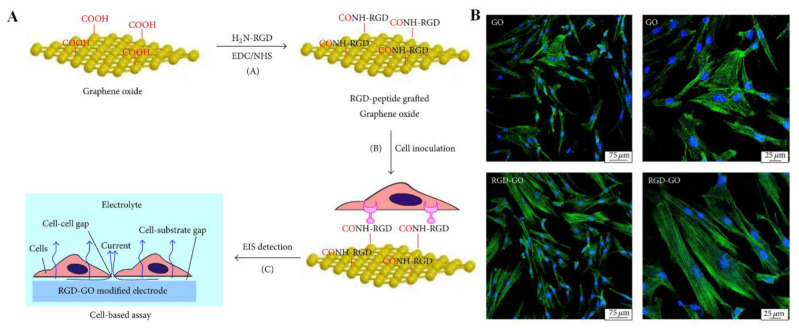
(**A**) Schematic illustration of the preparation of the RGD-GO film and the cell-based assay. (**B**) Fluorescence microscopy images of HPLFs grown on GO (**top**) or RGD-GO (**bottom**) films. Blue from DAPI and green color fluorescence from GFP indicate the nucleus and cytoskeleton, respectively. Reproduced with permission from Ref. [26]. Copyright © 2016 Jianxia Li et al., Hindawi.

**Figure 4 pharmaceutics-15-00345-f004:**
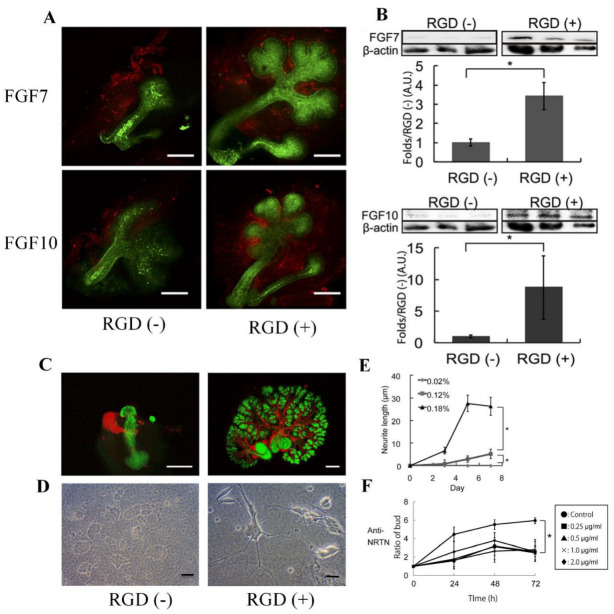
(**A**) Flow cytometric analysis of FGF7 and FGF10 expression in SMG tissue cultured for 24 h on hydrogel sheets with or without RGD modification (red—anti-FGF7/anti-FGF10, scale bar = 100 μm). (**B**) Western blot analysis of FGF7/FGF10 levels and β-actin as a control in SMG tissue cultured for 72 h. (**C**) Growth of neural networks in SMG tissue cultured for 72 h on hydrogel sheets (red—anti-βIII tubulin, green—PNA, scale bar = 100 μm). (**D**) PC12 neural cells cultured for 7 days on hydrogel sheets with and without RGD (scale bar = 100 μm). (**E**) Neurite growth of PC12 cells cultured on hydrogel sheets with and without RGD. (**F**) SMG growth on the RGD-modified sheet in the presence of increasing amounts of an anti-NRTN antibody (* *p* < 0.05). Reproduced with permission from Ref. [29]. Copyright © 2015, Taketa et al., Springer Nature.

**Figure 5 pharmaceutics-15-00345-f005:**
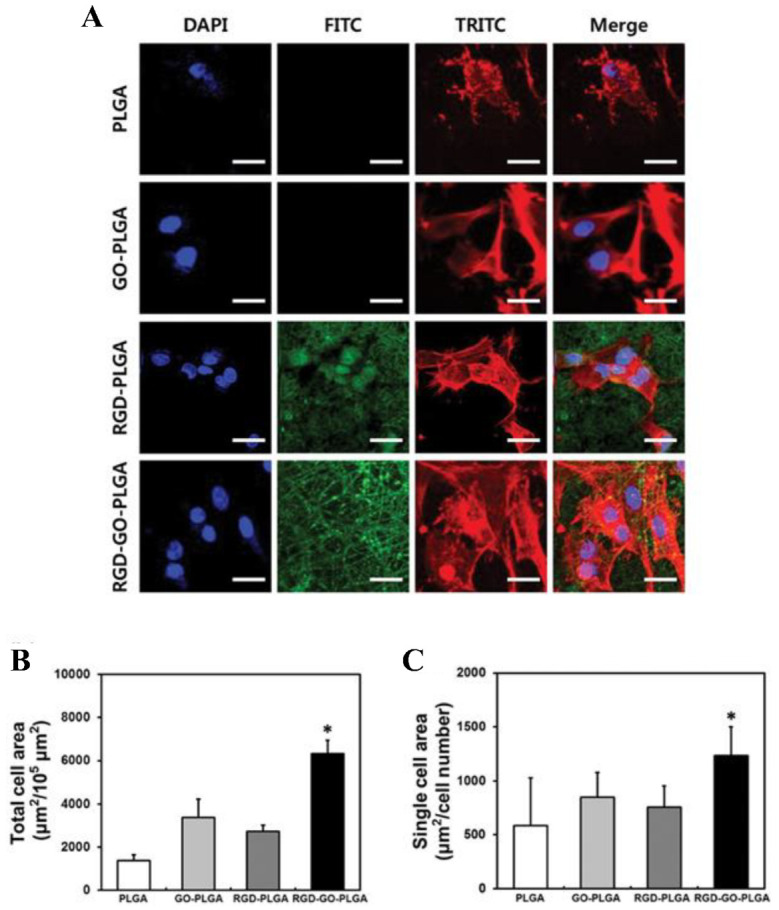
(**A**) Images of VSMCs on nanofiber mats composed of PLGA, GO-PLGA, RGD-PLGA, and RGD-GO-PLGA, as indicated. The cytoskeleton was stained with TRITC-labeled phalloidin (red), the nuclei with DAPI (blue), and the RGD-M13 phage in the nanofiber mats with a FITC-labeled antibody (green). Scale bar = 25 µm. (**B**) Quantification of the total cell area and (**C**) the single-cell area of the VSMC after three days of culture. Single-cell area was calculated by dividing the total cell area by the total number of nuclei (* *p* < 0.05). Reproduced with permission from Ref. [14]. Copyright © 2017, Oxford University Press.

**Figure 6 pharmaceutics-15-00345-f006:**
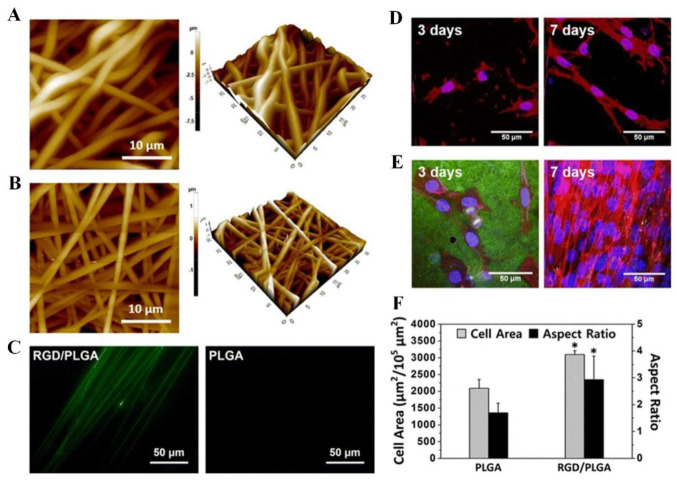
(**A**–**C**) Electrospun nanofiber surface morphology and immunostaining. (**A**) PLGA and (**B**) RGD/PLGA nanofiber matrix images obtained by atomic force microscopy. (**C**) PLGA and RGD/PLGA electrospun fiber immunofluorescence images following staining with FITC-labeled anti-M13 phage antibodies (green). (**D**–**F**) The morphologies of C2C12 myoblasts cultured on PLGA and RGD/PLGA nanofiber substrates. C2C12 myoblasts were cultured for three and seven days on (**D**) PLGA and (**E**) RGD/PLGA nanofiber matrices. Nuclei of the cells were counterstained with DAPI (blue), F-actins were stained with TRITC-labelled phalloidin (red), and RGD-M13 phages were immune-stained with FITC-labelled anti-M13 phage antibodies (green). (**F**) Quantification of the area and aspect ratio of the cells after three days (* *p* < 0.05). Reproduced with permission from Ref. [30]. Copyright © 2015, Shin et al., Springer Nature.

**Table 1 pharmaceutics-15-00345-t001:** RGD peptide-based and RGD-functionalized materials for TE applications.

Type of RGD Peptide	Structure of the RGD	Type of Tissue	Applications	References
RGD-GO-PLGA	Nanofibers	Muscle cells	Regeneration of vascular smooth muscle	[14]
Functionalization of chitosan, gelatin, β-glycerphosphate with RGD peptide	Hydrogel	Myocardial tissue	Vascularization and tissue regeneration	[76]
Chitosan/biphasic calcium phosphate scaffolds functionalized-RGD	Hydrogel	Bone tissue	Bone regeneration	[77]
RGD	Hydrogels	Bone tissue	Growth-factor-free microenvironment	[49]
c-RGDfK	Hydrogel	Bone tissue	Bone regeneration	[50]
Functionalized RGD	Hydrogels	Bone tissue	Bone sialoprotein	[55]
RGD-conjugated PEG-diacrylate (PEODA)	Hydrogel	Bone tissue	Osteogenesis of bone marrow derived stromal cells	[78]
RGD-peptide-coupled alginates	Gel	Bone tissue	Chemoenzymatic	[15]
RGD peptides	Gel	Bone tissue	Mesenchymal stem cell	[79]
RGD-modified three-dimensional porous PCL scaffolds	Gel	Bone tissue	Interaction between bone marrow stromal cells	[80]
RGD-modified silk	Fiber	Bone tissue	Ligament fibroblast responses	[81]
Lysine-cyclic RGD peptide	NA	Bone tissue	Cell retention technology	[21]
Laser groove/RGD-functionalized Ti-6Al-4V	NA	Bone tissue	Pins in rabbit femurs	[60]
RGD-decorated PLGA	Fiber	Skeletal tissue	Skeletal tissue regeneration	[30]
Hyaluronic acid/RGD-functionalized	Hydrogel	Cartilage tissue	Enhancement in chondrogenesis	[82]
Cyclic RGD-peptides	NA	Cartilage tissue	Articular cartilage	[16]
RGD	Fiber	Cardiac tissue	Cardiac tissue engineering	[22]
RGD motifs	Myofibers	Heart/cardiac tissue	Myocardial repair	[23]
RGDfK-Peptide	Gel	Cardiac tissue	Cell transplantation and cardiac neovascularization	[24]
Immobilized RGD peptide	Gel	Cardiac tissue	Cardiac tissue engineering	[25]
RGD-modified acellular bovine pericardium	Fiber	Acellular tissues	Scaffold for tissue engineered heart valves	[83]
RGD dimer peptides	Fiber	Aortic arch tissue	Imaging of high-risk atherosclerotic plaques	[84]
Cyclic RGD peptide	Hydrogels	Vascular grafts tissue	Enhancement in graft performance	[64]
RGD-modified poly(ester-urethane) urea	Fiber	Vascular tissue	Potential application for vascular tissue engineering	[67]
Functionalized RGD	Fiber	Endothelial tissue	Rabbit carotid artery model	[65]
RGD-functionalized polymer	Fibers	Endothelial cells	Adhesion of human umbilical vein endothelial cells	[20]
GRGDSP	NA	Blood tissue	Endothelization of small-diameter vascular grafts and tissue engineering of blood vessels	[32]
Recombinant RGD-Silk	Fiber	Human corneal cells	Substrates for human corneal cells	[70]
ECM peptide RGD	Fiber	Human corneal epithelial tissue	Corneal epithelial contact guidance	[85]
RGD-alginate	Hydrogel	Retinal tissue	Improvement in retinal tissue development	[74]
Helicoidal multi-lamellar features of RGD-functionalized	NA	Corneal tissue	Corneal tissue engineering	[86]
Bicyclic RGD peptides	Fiber	Neurites	Nerve growth	[48]
RGD-modified peptide	Hydrogel	Gland tissue	Gland tissue growth and morphology in vitro	[29]
RGD-modified polymers	Fiber	CNS tissue replacement	Stimulated cell adhesion	[17]
Functionalized RGD	Hydrogel	Neural tissue	Neural tissue repair	[45]
Phospholipid bilayers functionalized RGD peptides	3D Gel	Scaffold-based tissue	Neural stem cell adhesion and proliferation	[87]
Bicyclic RGD peptides	NA	NA	Integrin αvβ3 receptor	[19]
RGD	NA	Fibroactin tissue	Cell adhesion	[88]
RGD-modified recombinant spider silk proteins	Fiber	Fibroblasts	Cell adhesion and proliferation	[89]

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
