# Peer review of "Design of Functional RGD Peptide-Based Biomaterials for Tissue Engineering"

_pharmaceutics, 2023, doi:10.3390/pharmaceutics15020345_

Round 1

Reviewer 1 Report

The manuscript titled 'Design of Functional RGD Peptide-Based Biomaterials for Tissue Engineering' compiles recent literature relating to the application of RGD peptides for synthesizing biomaterials to foster the growth and proliferation of tissues. The literature has been surveyed fairly extensively and recent advances in the field have been categorized and distinguished systematically. Overall, the writing of the manuscript is easy to follow and represents an important compilation of relevant data. The few concerns associated with the manuscript have been listed below:

1) The authors have not clearly explored to what extent can cell fate and function be controlled by using RGD-based biomaterials compared to other peptide sequences across cell and tissue types.

2) It might be important to highlight screening strategies or methods currently employed to identify the optimal cell-adhesion peptides for a specific cell or tissue type.

3) Does employing the cell-adhesion RGD peptide help reduce the use of growth factors currently used to control cell/tissue spreading, differentiation, proliferation, survival and organization? Some recent literature has highlighted that integrin receptors that bind to RGD peptides are directly activated and regulated by growth factor receptor activation as they regulate mechanical and biophysical outcomes regulating tissue functions. See reference: J Cell Sci (2020) 133 (13): jcs238840. It might be worthwhile to have a discussion RGD associating integrin receptors and their functions when regulated by growth factors as these are also included in the formulation of biomaterials.

4) The manuscript includes a bunch of figures reproduced from previous publications. At times, this seems to be an overkill and the review loses its value as a stand-alone or perspective article. The authors are advised to provide figures which represent their interpretation of the data either as summarized models or figures if possible and reduced the total amount of reproduction. For this, the authors could also combine results from multiple papers and present proposed models. 

5) Lines 75-80 seem like instructions from a previous round of review or suggestions from the author's internal discussions. This should be removed as it is clearly misplaced within the manuscript. 

6) Figure 1 is difficult to comprehend and has some grammatical errors.  

Author Response

English language and style

( ) English very difficult to understand/incomprehensible
( ) Extensive editing of English language and style required
( ) Moderate English changes required
(x) English language and style are fine/minor spell check required
( ) I don't feel qualified to judge about the English language and style

Response: The English and typo errors has been corrected throughout the manuscript.

The manuscript titled 'Design of Functional RGD Peptide-Based Biomaterials for Tissue Engineering' compiles recent literature relating to the application of RGD peptides for synthesizing biomaterials to foster the growth and proliferation of tissues. The literature has been surveyed fairly extensively and recent advances in the field have been categorized and distinguished systematically. Overall, the writing of the manuscript is easy to follow and represents an important compilation of relevant data. The few concerns associated with the manuscript have been listed below:

The authors have not clearly explored to what extent can cell fate and function be controlled by using RGD-based biomaterials compared to other peptide sequences across cell and tissue types.

Response: We would like to thank the reviewer for taking the time and effort to review our manuscript and for the positive evaluation. The suggested notion has been added into the manuscript, as follows:

Page 10: Various RGD peptide sequences and hydrogels with varying structures have been used for tissue engineering applications. An RGD peptide can be classified as linear or circular. It is interesting to note that cyclic RGD peptides are believed to be more active than linear RGD peptides. Cyclic peptides likely have a higher affinity for integrin receptors and are more resistant to proteolysis. Additionally, a study by Li et al found that cyclic RGDs was more beneficial to bone repair in vivo than linear RGD[31]. The modification of poly-carbonate urethane with a photoactive peptide sequence containing RGD also resulted in improved vascular grafts[32]–[34]. Moreover, cell adhesion depends on mechanical forces, growth factors, and the ECM[35]. Several physical, chemical, and functional interactions between the cell, growth factors, and the structure and rigidity of the surrounding matrix determine the cell fate or adhesion[35]–[37]. The interaction between EGFR (epidermal growth factor receptor) and integrins has been demonstrated, however most of the observed outcomes have been attributed to molecules downstream of the receptors, away from the plasma membrane[36], [37]. In a recent study, Rao et al used tension gauge tether probes to display the integrin ligand cRGDfK and to measure the tension between the integrins[38]. The results of this study suggest that EGFR regulates integrin tension as well as the spatial organization of focal adhesions[38]. Additionally, the researchers found that the mechanical tension threshold required for outside-in integrin activation can be adjusted by ligand-dependent EGFR signaling.

2) It might be important to highlight screening strategies or methods currently employed to identify the optimal cell-adhesion peptides for a specific cell or tissue type.

Response: We thank the reviewer for the valuable suggestion. The information regarding identification of optimal cell-adhesion peptides and their applications has been added to the manuscript, as follows:

Page 3: The use of functionalized RGD materials with short cell-adhesion peptides (CAPs) has led to a better understanding of the role of the ECM. It is possible to easily translate RGD materials that have been functionalized with CAP into robust 3D cell culture systems for tissue engineering applications. Unfortunately, the panel of CAPs available for designing cell culture environments is so far limited to non-modified RGDs. A high-throughput screening approach has been proposed in order to evaluate the role of CAPs as well as identify their precise impact on cell adhesion [ref.]. In spite of the fact that microarray screening platforms have been established for studying cell adhesion, high-throughput screening has yet to be applied to tissue engineering and 3D cell culture. It is important to understand the role of CAPs in tissue engineering and design, not only for their cell-adhesion properties but also to drive cellular functions and organization.

3) Does employing the cell-adhesion RGD peptide help reduce the use of growth factors currently used to control cell/tissue spreading, differentiation, proliferation, survival and organization? Some recent literature has highlighted that integrin receptors that bind to RGD peptides are directly activated and regulated by growth factor receptor activation as they regulate mechanical and biophysical outcomes regulating tissue functions. See reference: J Cell Sci (2020) 133 (13): jcs238840. It might be worthwhile to have a discussion RGD associating integrin receptors and their functions when regulated by growth factors as these are also included in the formulation of biomaterials.

Response: We thank the reviewer for raising this issue. The suggested information has been added to the revised manuscript as follows:

Page 10: Moreover, cell adhesion depends on mechanical forces, growth factors, and the extracellular matrix[35]. To determine cell fate or cell adhesion, several physical, chemical, and functional relationships exist between a cell, growth factors, and the structure and rigidity of the surrounding matrix[35]–[37]. The interaction between EGFR (epidermal growth factor receptor) and integrins has been demonstrated in studies, however most of the observed outcomes have been attributed to molecules downstream of the receptors, away from the plasma membrane[36], [37]. In a recent study, Rao et al used tension gauge tether probes to display the integrin ligand cRGDfK and to measure the tension between the integrins[38]. The results of this study suggest that EGFR regulates integrin tension as well as the spatial organization of focal adhesions[38]. Additionally, the researchers found that the mechanical tension threshold required for outside-in integrin activation can be adjusted by ligand-dependent EGFR signaling.

4) The manuscript includes a bunch of figures reproduced from previous publications. At times, this seems to be an overkill and the review loses its value as a stand-alone or perspective article. The authors are advised to provide figures which represent their interpretation of the data either as summarized models or figures if possible and reduced the total amount of reproduction. For this, the authors could also combine results from multiple papers and present proposed models. 

Response: The total number of figures has been reduced and several paragraphs have been rephrased.

5) Lines 75-80 seem like instructions from a previous round of review or suggestions from the author's internal discussions. This should be removed as it is clearly misplaced within the manuscript. 

Response: The suggested sentences have been removed.

6) Figure 1 is difficult to comprehend and has some grammatical errors

Response: Figure 1 has been revised and corrected.  

Reviewer 2 Report

The submitted manuscript summarized functional RGD peptide-based biomaterials for tissue engineering. This topic is of interest for readers of Pharmaceutics. However, I have some reservations about the information provided. I therefore recommend publication of this manuscript only if the authors can address the major issues noted below.

1. The writing needs to be improved as some statements are quite general. E.g. “One of the most fundamental levels of organization of the human body is the tissue one.” The authors should increase the writing accuracy and highlight the significance clearly.

2. The authors described different types of RGD peptides. However, the examples are not well organised so that the readers can give a full picture of their functions in tissue engineering. I suggest the authors write a summary section with graphs or tables for different RGD peptides.

3. Figure 1 is a bit difficult to follow and please present in a more meaningful way or add more description in the figure caption.

3. The authors stated that “The pre-clinical applications and clinical trials of RGD-modified systems for different applications are briefly presented in Table 1.” It is not clear which examples are preclinical studies or clinical trials. This needs to be modified and make it clear.

4. According to the title of this review, it is assumed to focus on the role of solute carrier transporters in anticancer drug delivery. However, in the last part of this review (5.3.2), “Drug delivery via transporters highly expressed in cancer cells” is not the main section in this review, compared to other sections. Can consider editing the title to make it more accurate.

5. In general, this review is short of the reviewers' insights and in-depth summary. Particularly, the last section of the review should be improved and give more information about future trends.

6. Please also correct some typos and small errors.

Author Response

English language and style

( ) English very difficult to understand/incomprehensible
( ) Extensive editing of English language and style required
( ) Moderate English changes required
(x) English language and style are fine/minor spell check required
( ) I don't feel qualified to judge about the English language and style

Response: Throughout the manuscript, English and typographical errors have been corrected.  

Comments and Suggestions for Authors

The submitted manuscript summarized functional RGD peptide-based biomaterials for tissue engineering. This topic is of interest for readers of Pharmaceutics. However, I have some reservations about the information provided. I therefore recommend publication of this manuscript only if the authors can address the major issues noted below.

  1. The writing needs to be improved as some statements are quite general. E.g. “One of the most fundamental levels of organization of the human body is the tissue one.” The authors should increase the writing accuracy and highlight the significance clearly.

Response: We would like to thank the reviewer for taking the time and effort to review our manuscript and for the positive evaluation. The entire opening paragraph has been rephrased to better clarify our notions, as follows:

Page 2: The tissue level of organization is one of the most fundamental in the human body. Certain tissues have the remarkable ability to regenerate, which can be harnessed as a treatment for a wide range of injuries, such as fractured bones, cuts in muscles, and damage to organs that are otherwise difficult to treat[1], [2]. The regeneration of tissues is not always successful in case of diseases and injuries, raising the need for surgical intervention. Furthermore, the treatment and repair of extensive tissue damage may require tissue/cell grafts, replacements, or implants[1]. For this purpose, several strategies have been developed, such as autogenic and allogenic tissue implantation[3]. Although such implants have several benefits, they also entail some drawbacks, including insufficient donor organs and lifelong immunosuppression[3]. Furthermore, transplanted organs often cannot function as native ones[4], [5]. Aiming to address these issues, a variety of synthetic and natural materials have been utilized for tissue engineering (TE) and tissue implants. Importantly, TE is intended to not only restore damaged tissues but also to assemble functional constructs that promote regeneration and enhance cell viability and migration. Therefore, an implantable material must be biocompatible, biodegradable, and have proper functional sites that can adapt to biodegradation. Importantly, biomedical materials must be able to interact with cells in vivo to avoid any potential foreign body reactions such as implant encapsulation, thrombosis, embolization, and aseptic loosening. Unlike metals and ceramics, which are non-biodegradable and difficult to process, peptides show high design flexibility due to their structure and composition, allowing their facile design to support cell viability and proliferation[6]–[8]. 

  1. The authors described different types of RGD peptides. However, the examples are not well organised so that the readers can give a full picture of their functions in tissue engineering. I suggest the authors write a summary section with graphs or tables for different RGD peptides.

Response: The summary of the work has been modified as suggested.  

  1. Figure 1 is a bit difficult to follow and please present in a more meaningful way or add more description in the figure caption.

Response: Figure 1 has been revised and corrected.    

  1. The authors stated that “The pre-clinical applications and clinical trials of RGD-modified systems for different applications are briefly presented in Table 1.” It is not clear which examples are preclinical studies or clinical trials. This needs to be modified and make it clear.

Response: We have changed the context of the Table 1 and modified systematically.

  1. According to the title of this review, it is assumed to focus on the role of solute carrier transporters in anticancer drug delivery. However, in the last part of this review (5.3.2), “Drug delivery via transporters highly expressed in cancer cells” is not the main section in this review, compared to other sections. Can consider editing the title to make it more accurate.

Response: This suggestion is not relevant to the current manuscript, since there is no mention of cancer in this paper.

  1. In general, this review is short of the reviewers' insights and in-depth summary. Particularly, the last section of the review should be improved and give more information about future trends.

Response: The last section and summary has been revised.  

  1. Please also correct some typos and small errors.

Response: Corrections have been made to English and typographical errors throughout the manuscript.  

Round 2

Reviewer 2 Report

The authors have addressed my comments. I am happy to endorse this paper.